# THERMALGAUSSIAN: THERMAL 3D GAUSSIAN SPLATTING

**Rongfeng Lu**[1,3,*,†] **Hangyu Chen**[1,*] **Zunjie Zhu**[1,3] **Yuhang Qin**[1] **Ming Lu**[2] **Le Zhang**[1] **Chenggang Yan**[1] **Anke Xue**[1,†]
[1]Hangzhou Dianzi University  [2]Intel Labs China  [3]Lishui Institute of Hangzhou Dianzi University

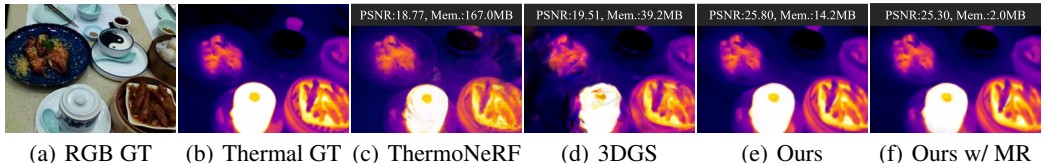

| (a) RGB GT | (b) Thermal GT | (c) ThermoNeRF | (d) 3DGS | (e) Ours | (f) Ours w/ MR |

Figure 1: Compared to NeRF-based methods (Hassan et al., 2024) and methods that directly use thermal images for training 3DGS, our approach not only improves the thermal image rendering quality but also significantly reduces the model's storage through multimodal regularization (MR).

## ABSTRACT

Thermography is especially valuable for the military and other users of surveillance cameras. Some recent methods based on Neural Radiance Fields (NeRF) are proposed to reconstruct the thermal scenes in 3D from a set of thermal and RGB images. However, unlike NeRF, 3D Gaussian splatting (3DGS) prevails due to its rapid training and real-time rendering. In this work, we propose ThermalGaussian, the first thermal 3DGS approach capable of rendering high-quality images in RGB and thermal modalities. We first calibrate the RGB camera and the thermal camera to ensure that both modalities are accurately aligned. Subsequently, we use the registered images to learn the multimodal 3D Gaussians. To prevent the overfitting of any single modality, we introduce several multimodal regularization constraints. We also develop smoothing constraints tailored to the physical characteristics of the thermal modality. Besides, we contribute a real-world dataset named RGBT-Scenes, captured by a hand-hold thermal-infrared camera, facilitating future research on thermal scene reconstruction. We conduct comprehensive experiments to show that ThermalGaussian achieves photorealistic rendering of thermal images and improves the rendering quality of RGB images. With the proposed multimodal regularization constraints, we also reduced the model's storage cost by 90%. Our project page is at https://thermalgaussian.github.io/.

## 1 INTRODUCTION

Thermal imaging is widely used in fields such as military (He et al., 2021), healthcare (Lahiri et al., 2012), industry (Glowacz, 2021), agriculture (Zhou et al., 2021), building inspection (El Masri & Rakha, 2020), and search and rescue (Yeom, 2024) because it converts temperature information—an important physical modality not visible to the human eye—into interpretable images. 3D reconstruction technology, which involves lifting multi-view 2D images into 3D scenes, is foundational for key technologies such as the metaverse, digital twins, autonomous driving, and robotics. Any image with valuable 2D applications can be lifted into 3D to view the captured scene from a new view and in greater detail. Thermal images are no exception (Abreu de Souza et al., 2023; Liu et al., 2024).

Previous 3D thermal scene reconstruction (Rangel et al., 2014; Zhao et al., 2017; Müller, 2019; Li et al., 2023) involves a two-stage process. In the first stage, RGB images and traditional multi-view

---

*Equal contribution
†Correspondence to: rongfeng-lu@hdu.edu.cn, akxue@hdu.edu.cn

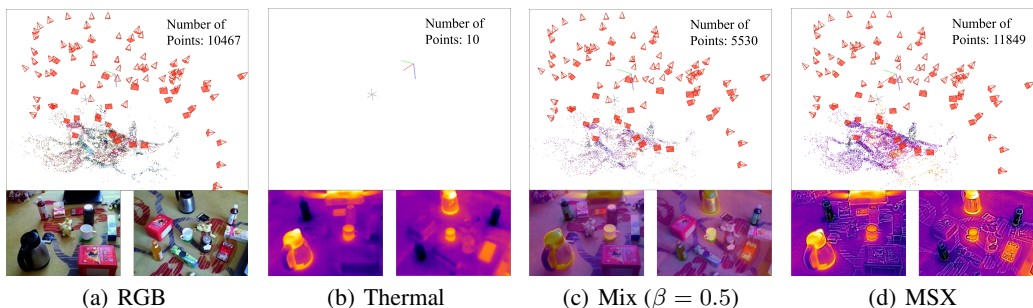

Figure 2: Top: camera poses and point cloud generated by SfM. Bottom: input images for SfM.

geometry methods (Newcombe et al., 2011) are used to achieve a 3D geometric reconstruction. In the second stage, thermal images are mapped onto the reconstructed 3D scene. However, these methods not only fail to fully exploit thermal information but are also constrained by the limitations of traditional 3D reconstruction techniques, which impede their ability to render high-quality images from a new view. This hinders the practical application of thermal reconstruction.

Neural Radiance Fields (NeRF) (Mildenhall et al., 2021) can render photorealistic images from a new view, thus revolutionizing novel-view synthesis and 3D reconstruction. Recently, several NeRF-based methods (Hassan et al., 2024; Ye et al., 2024) have been proposed to reconstruct thermal scenes in 3D using thermal images. However, NeRF's slow rendering speed and implicit scene representation limit its practical applications. In contrast, 3D Gaussian Splitting (3DGS) (Kerbl et al., 2023) not only maintains photorealistic rendering quality from new views but also significantly improves rendering speed. Additionally, the explicit 3D scene representation of 3DGS, akin to point clouds, facilitates integration with downstream tasks, establishing it as a leading approach in research. Building on 3DGS, we propose ThermalGaussian, a multimodal Gaussian technique that renders high-quality RGB and thermal images from new views.

The absence of open-source datasets dedicated to thermal scene reconstruction significantly impedes progress in this domain. Several researchers (Hassan et al., 2024; Ye et al., 2024) have recognized this issue and have released some datasets. However, these datasets suffer from problems such as lack of color images registered with thermal images, inconsistencies in thermal information from different views, and watermarked images. To address these issues, we contribute a real-world dataset named RGBT-Scenes.

Unlike RGB images, thermal images possess unique low-texture and ghosting characteristics (Bao et al., 2023) that hinder accurate camera pose estimation using Structure-from-Motion (SfM) (Schönberger & Frahm, 2016), as illustrated in Fig. 2b. Consequently, thermal images cannot directly replace RGB images for running 3DGS. To address this issue, we first register the RGB and thermal images and then fuse them (Fig. 2c), or use Multi-Spectral Dynamic Imaging (MSX) (Abdullah, 2023) (Fig. 2d) to localize the thermal image camera. Additionally, We design a thermal loss to adapt to the unique characteristics of thermal images.

Introducing a new modality, such as thermal imaging, into 3D reconstruction should enable the model to understand the scene from a more comprehensive perspective. However, ThermoNeRF (Hassan et al., 2024) reduces the RGB rendering quality after implementing thermal reconstruction. In contrast, our method not only improves thermal rendering quality but also enhances RGB rendering quality by 1 dB. Furthermore, to prevent overfitting of any single modality during multimodal Gaussian training, we introduce a multimodal regularization coefficient. This approach significantly reduces model storage requirements and accelerates rendering speed. In summary, the main contributions as follows:

(1)We propose ThermalGaussian, the first multimodal 3DGS capable of simultaneously rendering photorealistic thermal and RGB images of a scene.

(2)We propose a series of strategies for multimodal Gaussian reconstruction, including multimodal initialization, three different thermal Gaussians, constraints specific to thermal modalities, and multimodal regularization.

(3)We introduce RGBT-Scenes, a new dataset designed for thermal 3D reconstruction and novel-view synthesis. The dataset consists of paired RGB and thermal images captured from multiple viewpoints across 10 different scenes.

(4)Finally, experimental results show that our multimodal method not only improves the rendering quality of both thermal and RGB images but also reduces storage space by 90% compared to training each modality separately, while also improving rendering speed.

## 2 RELATED WORK

### 2.1 THERMAL IMAGING AND 3D RECONSTRUCTION

All objects with temperatures above absolute zero emit energy in the form of electromagnetic waves, a phenomenon known as thermal radiation. Through Planck's law, thermal imaging converts the wavelength and intensity of electromagnetic waves radiated from an object's surface into thermal information, which is then used to create images. Common thermal imaging devices operate in the mid-to-long-wave infrared range. To intuitively display the temperature distribution in thermal imaging, pseudo-color rendering is often applied using a cool-to-warm color scheme. Originally, thermal imaging was developed for military purposes to enable visualization under extreme lighting conditions, such as nighttime or smoke. As costs have decreased in recent years, it has also been widely applied in fields of healthcare, industry, agriculture, building inspection, and so on.

By capturing 2D images from various angles and applying 3D reconstruction, a 3D model can be created. Unlike fixed 2D viewpoints, the 3D model enables intuitive analysis from any perspective and scale, allowing detailed exploration. The advent of KinectFusion (Newcombe et al., 2011) marked the beginning of an era of high-precision, dense 3D reconstruction. Subsequent developments (Nießner et al., 2013; Kähler et al., 2015; Dai et al., 2017; Gong et al., 2021; Zhang et al., 2021) have optimized 3D reconstruction in terms of accuracy, efficiency, and unconstrained camera movement. Some works (Rangel et al., 2014; Zhao et al., 2017; Müller, 2019; Li et al., 2023) have been made to integrate thermal imaging with the aforementioned methods, leading to the development of thermal 3D reconstruction. However, these traditional multi-view geometry-based methods do not perform as well in rendering new views as the more recent deep learning-based approaches.

With the rapid development of artificial intelligence(Yan et al., 2020a; Wang et al., 2024a), NeRF (Mildenhall et al., 2021) has emerged as a significant milestone in the field of 3D reconstruction due to its impressive ability to render highly realistic images from a new view. Recently, ThermoNeRF (Hassan et al., 2024) and Thermal-NeRF (Ye et al., 2024) have been proposed to reconstruct thermal scenes by combining a set of thermal images with NeRF. Although these approaches successfully generate images from new perspectives, they are limited by the slow rendering speed and implicit scene representation of NeRF, which hampers their practical application.

### 2.2 3DGS AND MULTIMODALITY

3DGS (Kerbl et al., 2023) represent a revolutionary technology in the fields of 3D reconstruction. Distinct from methods like NeRF, 3DGS employs millions of explicit Gaussians, fundamentally altering its approach. This technology merges the advantages of neural network-based optimization with structured data representation, enabling photorealistic rendering from new views, significantly enhancing real-time rendering capabilities, and introducing the ability to manipulate and edit 3D scenes. These features make 3DGS highly compatible with a broad range of downstream applications, establishing it as the baseline for next-generation 3D reconstruction technologies (Chen & Wang, 2024). Although 3DGS is constrained and trained using only RGB modality images, it ultimately generates millions of Gaussians, resembling a point cloud. This characteristic makes 3DGS particularly suitable for multimodal fusion with other devices that directly capture scene point clouds, such as depth cameras and LiDAR. Studies (Matsuki et al., 2024; Yan et al., 2024; Keetha et al., 2024) have effectively integrated depth cameras to implement 3DGS-based simultaneous localization and mapping. Studies (Li et al., 2024; Chung et al., 2024) have effectively combined depth images (Yan et al., 2020b) estimated from a pre-trained Monocular Depth estimation model (Bhat et al., 2023) with the RGB modality, resulting in improved rendering quality and more accurate geometric structures. 3D scenes contain not only RGB and geometric modalities but also other important modalities relevant to various applications, such as thermal, material, and pressure.

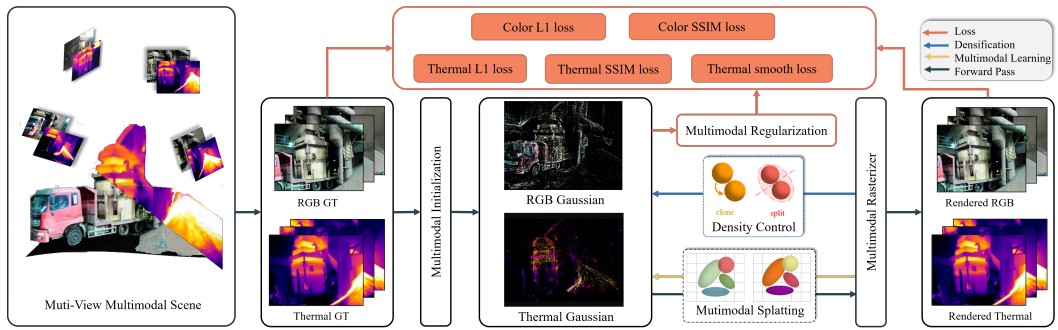

Figure 3: **ThermalGaussian Overview.** We simultaneously construct Gaussians for RGB and thermal modalities using the point cloud obtained from multimodal initialization. Each modality's Gaussians are used to render images in their respective modality. However, the losses from different modalities are combined to jointly constrain the optimization of both sets of Gaussians. Additionally, we establish a multimodal regularization based on the number of Gaussians in each modality, which dynamically adjusts the training coefficients for both modalities.

## 3 METHOD

Fig. 3 shows the overview of the proposed ThermalGaussian, which is based on the 3DGS (Kerbl et al., 2023), aiming to extend its capability to simultaneously render images of color and temperature. In this section, we first briefly introduce the background of the 3DGS. Then, we provide a detailed description of our method's specific implementation details, including multimodal initialization, three types of multimodal thermal Gaussians, thermal loss, and multimodal regularization.

### 3.1 PRELIMINARY: 3D GAUSSIAN SPLATTING

3DGS (Kerbl et al., 2023) represents a 3D reconstruction scene using a large number of anisotropic 3D Gaussians. This representation not only provides differentiability, which offers advantages in learning-based methods, but also enables explicit spatial expression, enhancing the editability and controllability of 3D scenes. Furthermore, it allows for rapid and efficient rasterization rendering through splatting. Initially, a set of unordered images of objects to be reconstructed is processed using SfM to obtain the camera poses and sparse point clouds. 3DGS then initializes these sparse point clouds as the position $\mu$ of a 3D Gaussian:

$$G(x) = e^{-\frac{1}{2}(x-\mu)^T \Sigma^{-1} (x-\mu)} \tag{1}$$

where $\Sigma$ represents the covariance matrix of the 3D Gaussian, and $x$ denotes any point in the 3D scene. $\Sigma$ is defined using a scaling matrix $\boldsymbol{S}$ and a rotation matrix $\boldsymbol{R}$:

$$\Sigma = \boldsymbol{R}\boldsymbol{S}\boldsymbol{S}^T\boldsymbol{R}^T \tag{2}$$

The 3D Gaussian $G(x)$ is projected onto the imaging plane using the camera's intrinsic parameters, transforming it into a 2D Gaussian. Subsequently, the image is rendered through alpha-blending:

$$C(x') = \sum_{k \in N} c_k \alpha_k \prod_{j=1}^{k-1} (1 - \alpha_j) \tag{3}$$

where $x'$ represents the queried pixel position, $N$ denotes the number of 2D Gaussians corresponding to this pixel, $\alpha$ denotes the opacity of each Gaussian and the color $c$ on each Gaussian is modeled spherical harmonics. All attributes of the 3D Gaussians are learnable and optimized directly in an end-to-end manner during training.

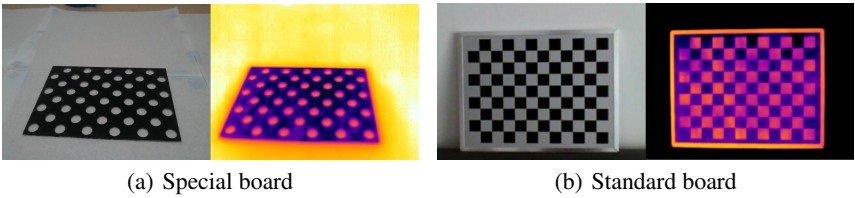

(a) Special board                     (b) Standard board

Figure 4: Different calibration boards for thermal Cameras.

## 3.2 MULTIMODAL INITIALIZATION

Previously, methods for calibration RGB and thermal images (Zhang et al., 2023) often involve designing specialized, non-standard metallic calibration boards with uniformly sized circular holes, as shown in Fig. 4a. The calibration relies on the temperature difference between the board and the background to compute thermal features, enabling calibration. However, the high complexity and stringent requirements for producing these calibration boards make them difficult to obtain and lack a universal standard. We find that a standard chessboard pattern, as shown in Fig 4b, commonly used for RGB camera calibration, can effectively be used for calibrating both thermal and color cameras, with a mean reprojection error of less than 0.5 pixels. Initially, we heat the calibration board using devices like an infrared heater; black regions, absorbing heat faster due to their material properties, exhibit relatively higher temperatures. We capture simultaneous color and thermal images before thermal equilibrium, which occurs when two systems reach a balanced state with equal temperatures, halting heat flow. Subsequently, conventional camera calibration (Zhang, 1999) is performed.

Using the calibrated intrinsic parameters $K_{\mathrm{RGB}}$ for the color camera, $K_{\mathrm{Th}}$ for the thermal camera, and the rotation $R$ and translation $t$ from the temperature camera to the color camera, we computed the corresponding positions $(u_{\mathrm{Th}}, v_{\mathrm{Th}})$ on the thermal image mapped to the registered positions on the color image:

$$\left[ \begin{array}{c} u_{\mathrm{RGB}} \\ v_{\mathrm{RGB}} \\ 1 \end{array} \right] = K_{\mathrm{RGB}} \left( R \cdot K_{\mathrm{Th}}^{-1} \left[ \begin{array}{c} u_{\mathrm{Th}} \\ v_{\mathrm{Th}} \end{array} \right] + t \right) \tag{4}$$

As shown in Fig.2(b), directly using thermal images, which exhibit low texture and ghosting characteristics, makes it difficult to successfully run SfM (Schonberger & Frahm, 2016). Therefore, to obtain the thermal camera poses, we tested three different multimodal SfM strategies. The first utilizes registered high-texture RGB images directly for camera pose estimation. These poses serve simultaneously for both the RGB and thermal cameras. However, practical scenarios that require thermal scene reconstruction often occur under dim lighting conditions or in scenes lacking distinct color features. Therefore, relying solely on color images may impede the precise camera pose estimation necessary for thermal scene reconstruction. The second approach, illustrated in Fig.2(c), involves blending registered color and thermal images using the following formula:

$$I_{\mathrm{mix}} = \beta I_{\mathrm{Th}} + (1 - \beta) I_{\mathrm{RGB}} \tag{5}$$

where in the above equation, $\beta$ represents the opacity of the thermal image. This method produces blended images containing both rich color and thermal information, catering to various practical applications of thermal scene reconstruction. The third strategy, depicted in Fig.2(d), maps high-frequency color variations from the color images onto the thermal images. This approach mitigates the lack of feature points caused by thermal images' low texture and ghosting characteristics.

## 3.3 THERMAL GAUSSIAN

We utilize three different multimodal training strategies to construct the thermal Gaussian.

**Multimodal Fine-Tuning Gaussians (MFTG):** Inspired by the fine-tuning approach used in large-scale models, our first multimodal training strategy is training a basic Gaussian with RGB images and then refining this Gaussian with thermal images to generate thermal Gaussian. This is a two-stage process. In the first stage, similar to 3DGS, we utilize multimodal camera poses and initial

point clouds obtained from multimodal initialization as inputs. The training is supervised using RGB images, with $\mathcal{L}_1$ combined with a D-SSIM term:

$$\mathcal{L}_{\text{RGB}} = (1 - \lambda)\mathcal{L}_1 + \lambda\mathcal{L}_{\text{D-SSIM}} \tag{6}$$

This stage enables us to render high-quality RGB modality images from a new view and establish a basic 3D Gaussian with preliminary geometry. In the second stage, we fine-tune this basic Gaussian model with thermal images and multimodal camera poses obtained from initialization. Since the first stage constraints were based on texture-rich color images rather than thermal images, which results in a better geometry. Therefore, training on this geometry yields better results than training thermal Gaussian directly from the initial point cloud derived from multimodal initialization.

**Multiple Single-Modal Gaussians (MSMG):** The training of MFTG initially utilizes RGB modal information followed by thermal modal information. Although both modalities are employed, they are not used simultaneously. Since only thermal images were utilized for supervision in stage two, the information from the color modality was not fully leveraged. Therefore, in MSMG (as shown in Fig. 3), we constrain the training with information from both color and thermal modalities simultaneously. We train two single-modal Gaussians initialized by point clouds from multimodal initialization. The thermal Gaussian renders thermal images, while the RGB Gaussian renders RGB images. Subsequently, these rendered images of both modalities are compared separately with the ground truth of their respective inputs using loss functions:

$$\mathcal{L} = \mathcal{L}_{\text{RGB}} + \mathcal{L}_{\text{thermal}} \tag{7}$$

The details of $\mathcal{L}_{thermal}$ constraint will be elaborated below. Each Gaussian model is influenced not only by its corresponding input modality but also by others. Experimental results indicate that joint constraints across multiple modalities enhance the training outcomes for both color and thermal modalities. Moreover, since these modalities jointly optimize the entire scene from different perspectives, redundant points are pruned to some extent, reducing the number of points in the point cloud and lowering the model's storage requirements.

**One Multi-Modal Gaussian(OMMG):** OMMG extends MSMG by not only employing dual-modal loss constraints in Eq. (7) but also integrating multiple modalities onto a single Gaussian. This integration ensures that information from diverse modalities is unified within a single geometric structure. Specifically, we construct a multimodal Gaussian comprising positional coordinates $x$, scaling matrix $\mathbf{S}$, rotation matrix $\mathbf{R}$, opacity $\alpha$, spherical harmonics $c$ for RGB representation, and spherical harmonics $t$ for thermal representation. RGB rendering is achieved using Formula 3, while thermal rendering follows the equation below:

$$T(x') = \sum_{k \in N} t_k \alpha_k \prod_{j=1}^{k-1} (1 - \alpha_j) \tag{8}$$

## 3.4 Thermal Loss & Multimodal Regularization

The loss function for RGB modality images is directly given by Equation 6. The same loss function can also be applied to thermal modality images. However, because thermal images exhibit unique low-texture and ghosting characteristics, we design a specific thermal loss function to better accommodate these features.

The RGB modality may exhibit abrupt changes. However, because all objects above absolute zero continuously engage in heat transfer and thermal radiation, eventually reaching thermal equilibrium with their surroundings, significant abrupt changes are typically not observed in thermal images. Additionally, most regions of objects in thermal equilibrium have similar temperatures, resulting in smoother thermal images. Therefore, we introduce a smoothness term for regularization:

$$\mathcal{L}_{\text{smooth}} = \frac{1}{4M} \sum_{i,j} (|T_{i\pm1,j} - T_{i,j}| + |T_{i,j\pm1} - T_{i,j}|) \tag{9}$$

where $T_{i,j}$ represents rendered thermal values at pixel position $(i, j)$. $M$ denotes the number of rendering pixels. Similarly to the color modality, we also incorporate $\mathcal{L}_1$ and $\mathcal{L}_{\text{D-SSIM}}$ Thus, our final temperature loss is:

$$\mathcal{L}_{\text{thermal}} = (1 - \lambda)\mathcal{L}_1 + \lambda\mathcal{L}_{\text{SSIM}} + \lambda_{\text{smooth}}\mathcal{L}_{\text{smooth}} \tag{10}$$

Table 1: Comparison of our collected dataset with others.

| Dataset | Mode T / RGB | Bimodal Calibration | Multiview Consistency | Content Richness |
|---|---|---|---|---|
| Thermal-NeRF | ✓ / × | × | - | Simple |
| ThermoNeRF | ✓ / ✓ | × | × | Moderate |
| Ours | ✓ / ✓ | ✓ | ✓ | Rich |

where $\lambda_{\text{smooth}}$ is the coefficient of $\mathcal{L}_{\text{smooth}}$.

When training multiple modalities of the same object simultaneously, it is often undesirable for one modality to dominate at the expense of others. Therefore, a regularization strategy is needed to dynamically adjust the weight of each modality's loss during training. In the training of MSMG, we observe that the weight coefficients of a modality align linearly with the Gaussian it ultimately generates. A higher weight for a modality results in more Gaussian generated by that modality, while fewer Gaussian are generated by another modality. Hence, we design multi-modal regularization coefficients $\gamma$ based on the number of Gaussian generated by each modality during training.

$$\gamma = \frac{N_{thermal}}{N_{thermal} + N_{RGB}} \tag{11}$$

where $N_{thermal}$ represents the number of Gaussian for the thermal modality during training. When the number of one modality's Gaussian increases, we increase the training weight of the other modality. This dynamic balancing of weights ultimately prevents overfitting to any single modality. The final design of this loss is:

$$\mathcal{L} = \gamma \mathcal{L}_{\text{RGB}} + (1 - \gamma)\mathcal{L}_{\text{thermal}} \tag{12}$$

## 4 SELF-COLLECTED THERAML DATASET

We introduce a new dataset, named RGBT-Scenes, which consists of aligned collections of thermal and RGB images captured from various viewpoints of a scene. The images are collected using the commercial-grade handheld thermal-infrared camera FLIR E6 PRO (Teledyne FLIR, 2024), which can simultaneously capture RGB and thermal images. The basic specifications of this camera include a resolution of 240×180, a field of view of 33°×25°, a temperature range from -20°C to 550°C, and a temperature accuracy of ±2% of the reading. Our dataset includes over 1,000 RGB and thermal images from 10 different scenes. These scenes encompass both indoor and outdoor environments, various object sizes (from large structures to everyday items), different temperature variations (ranging from a 300°C difference to a 4°C difference), and include both 360-degree and forward-facing scenarios. We provide the raw images captured by the thermal camera, as well as the RGB images, thermal images, MSX images, and camera pose data. In Table 1, we compare our dataset with those from concurrent works, Thermal-NeRF (Ye et al., 2024) and ThermoNeRF (Hassan et al., 2024). Our dataset includes both RGB and thermal images and applies multimodal calibration methods to align these images. The images used for calibration will also be made available. Compared to ThermoNeRF, our dataset ensures consistent thermal measurements across views and encompasses a richer variety of scenes. Detailed descriptions of each scene are provided in the supplementary.

## 5 EXPERIMENTS

### 5.1 IMPLEMENTATION DETAILS

Our method is an improvement upon the 3DGS framework, with all experimental settings (e.g., $\lambda$) remaining consistent with the reference 3DGS. The specific hyperparameter $\lambda_{smooth}$ is set to 0.6. Each comparative experiment was trained for 30K iterations. All experiments are conducted on a single NVIDIA 3090 GPU. The resolution of the rendered RGB images and thermal images is 640×480.

Table 2: Quantitative evaluation of thermal image using our method compared to previous work from test views. "×" indicates a failure to localize using only thermal images in the scene, making it impossible to success with 3DGS. 3DGS+MI represents the results obtained by directly training 3DGS after Multimodal Initialization.

| Metric | Method | Dimsum | Daily Stuff | Ebike | Road Block | Truck | Rotary Kiln | Building | Iron Ingot | Parterre | Land Scape | Avg. |
|---|---|---|---|---|---|---|---|---|---|---|---|---|
| PSNR ↑ | 3DGS | 25.38 | × | × | × | 20.97 | 23.79 | 23.75 | × | × | × | × |
| | ThermoNeRF | 24.27 | 17.34 | 19.70 | 17.17 | 23.53 | 26.40 | 23.31 | 22.97 | 17.88 | 18.79 | 21.13 |
| | 3DGS+MI | 26.35 | 18.77 | 20.89 | 26.75 | 26.17 | 26.59 | 25.76 | 29.57 | 22.09 | 20.17 | 24.31 |
| | Ours$_{MFTG}$ | **26.94** | 20.52 | 22.51 | 24.96 | 25.02 | 26.91 | 26.11 | **30.41** | 23.55 | 20.03 | 24.70 |
| | Ours$_{MSMG}$ | 26.73 | 21.35 | 23.23 | 26.52 | **26.27** | **27.15** | 26.83 | 30.06 | 25.01 | 20.61 | 25.38 |
| | Ours$_{OMMG}$ | 26.46 | **22.28** | **23.31** | **27.17** | 25.88 | 26.33 | 26.72 | 29.86 | **26.16** | **22.27** | **25.64** |
| SSIM ↑ | 3DGS | 0.860 | × | × | × | 0.717 | 0.872 | 0.810 | × | × | × | × |
| | ThermoNeRF | 0.747 | 0.759 | 0.694 | 0.781 | 0.750 | 0.916 | 0.804 | 0.717 | 0.709 | 0.774 | 0.765 |
| | 3DGS+MI | 0.889 | 0.789 | 0.806 | 0.917 | 0.872 | 0.922 | 0.872 | 0.887 | 0.843 | 0.794 | 0.859 |
| | Ours$_{MFTG}$ | **0.890** | 0.798 | 0.845 | 0.906 | **0.880** | 0.920 | 0.886 | 0.895 | 0.859 | 0.808 | 0.869 |
| | Ours$_{MSMG}$ | 0.891 | 0.829 | 0.857 | 0.909 | 0.879 | **0.926** | **0.897** | **0.898** | 0.860 | 0.832 | 0.878 |
| | Ours$_{OMMG}$ | 0.886 | **0.835** | **0.862** | **0.919** | 0.874 | 0.922 | 0.888 | 0.896 | **0.883** | **0.850** | **0.882** |
| LPIPS ↓ | 3DGS | 0.157 | × | × | × | 0.281 | 0.193 | 0.299 | × | × | × | × |
| | ThermoNeRF | 0.312 | 0.494 | 0.290 | 0.293 | 0.291 | 0.170 | 0.234 | 0.152 | 0.309 | 0.264 | 0.280 |
| | 3DGS+MI | 0.124 | 0.274 | 0.313 | 0.204 | 0.139 | 0.125 | 0.211 | 0.093 | 0.252 | 0.328 | 0.206 |
| | Ours$_{MFTG}$ | **0.121** | 0.258 | 0.235 | 0.210 | **0.133** | 0.129 | 0.199 | 0.091 | 0.232 | 0.317 | 0.192 |
| | Ours$_{MSMG}$ | 0.124 | **0.208** | 0.220 | 0.213 | **0.133** | 0.130 | 0.189 | **0.086** | 0.227 | 0.293 | 0.182 |
| | Ours$_{OMMG}$ | 0.129 | 0.210 | **0.203** | **0.198** | 0.136 | **0.124** | **0.177** | 0.091 | **0.181** | **0.248** | **0.170** |

## 5.2 THERMAL VIEW SYNTHESIS

Similar to 3DGS, we employ image quality assessment metrics including Peak Signal-to-Noise Ratio (PSNR) (Hore & Ziou, 2010; Wang et al., 2024b), Structural Similarity Metric (SSIM) (Wang et al., 2004), and Learned Perceptual Image Patch Similarity (LPIPS) (Zhang et al., 2018) to evaluate the quality of reconstructed thermal and RGB images from new views.

As shown in Table 2, even in scenes with pronounced thermal variations, specifically targeting low-texture thermal characteristics, direct application of thermal data proves challenging for 3DGS. In very few successful cases, inadequate precision in thermal camera positioning has compromised the quality of thermal reconstructions. 3DGS+MI denotes training the original 3DGS using thermal images instead of RGB images after obtaining accurate thermal poses through our multimodal initialization. Compared to 3DGS, 3DGS+MI adapts to a wider range of scenarios and achieves higher reconstruction quality. Given the higher reconstruction quality of 3DGS (Kerbl et al., 2023) com-

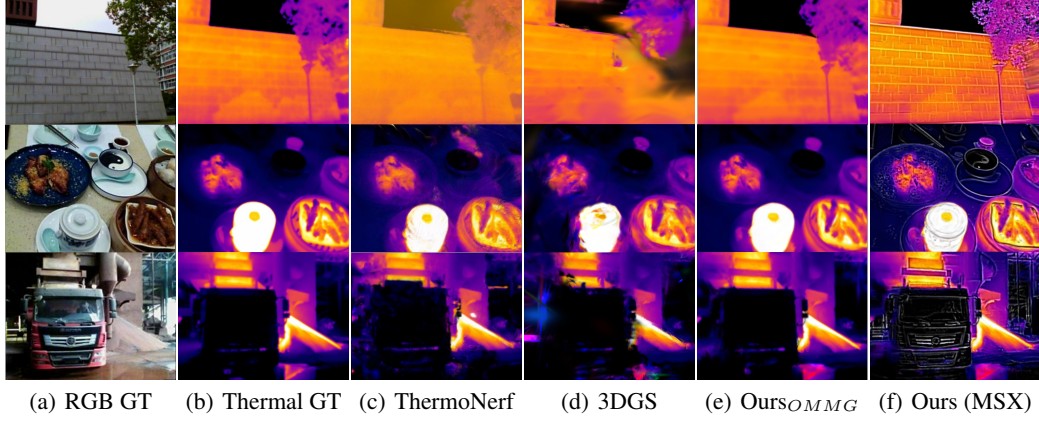

| (a) RGB GT | (b) Thermal GT | (c) ThermoNerf | (d) 3DGS | (e) Ours$_{OMMG}$ | (f) Ours (MSX) |

Figure 5: We present qualitative thermal image comparisons between our method, previous approaches (Hassan et al., 2024; Kerbl et al., 2023), and the corresponding ground truth images from test views. We also show the training results of the MSX images, which are easier to apply.

Table 3: Quantitative evaluation of RGB image using our method compared to 3DGS.

| Metric | Method | Dimsum | Daily Stuff | Ebike | Road Block | Truck | Rotary Kiln | Building | Iron Ingot | Parterre | Land Scape | Avg. |
|---|---|---|---|---|---|---|---|---|---|---|---|---|
| PSNR ↑ | 3DGS | 23.91 | 20.43 | 26.77 | 27.80 | 22.30 | 20.79 | 20.95 | 23.96 | 24.91 | 20.20 | 23.20 |
| | ThermoNeRF | 19.74 | 16.79 | 17.75 | 18.32 | 18.77 | 18.89 | 17.12 | 15.07 | 23.13 | 19.13 | 18.46 |
| | Ours$_{MSMG}$ | **24.42** | 21.71 | **27.34** | **28.22** | 23.57 | 22.23 | 23.08 | **25.69** | **25.57** | 20.91 | 24.27 |
| | Ours$_{OMMG}$ | 24.38 | **21.76** | 26.85 | 28.12 | **24.17** | **23.14** | **24.19** | 24.55 | 25.48 | **21.71** | **24.34** |
| SSIM ↑ | 3DGS | 0.847 | 0.748 | 0.901 | 0.910 | 0.807 | 0.772 | 0.791 | 0.872 | 0.859 | 0.696 | 0.820 |
| | ThermoNeRF | 0.688 | 0.639 | 0.540 | 0.619 | 0.688 | 0.600 | 0.460 | 0.293 | 0.756 | 0.583 | 0.586 |
| | Ours$_{MSMG}$ | **0.858** | 0.793 | **0.917** | 0.916 | 0.833 | 0.811 | 0.844 | **0.891** | **0.874** | 0.715 | 0.845 |
| | Ours$_{OMMG}$ | **0.858** | **0.797** | 0.905 | **0.920** | **0.840** | **0.822** | **0.849** | 0.884 | 0.855 | **0.739** | **0.847** |
| LPIPS ↓ | 3DGS | **0.194** | 0.299 | 0.171 | 0.201 | 0.232 | 0.217 | 0.228 | 0.188 | 0.183 | 0.280 | 0.219 |
| | ThermoNeRF | 0.228 | 0.465 | 0.244 | 0.548 | 0.311 | 0.207 | 0.291 | 0.301 | **0.167** | 0.275 | 0.303 |
| | Ours$_{MSMG}$ | **0.194** | 0.262 | **0.156** | 0.221 | 0.217 | 0.190 | **0.168** | **0.172** | 0.184 | 0.275 | 0.204 |
| | Ours$_{OMMG}$ | **0.194** | **0.253** | 0.169 | **0.199** | 0.211 | **0.184** | 0.170 | 0.186 | 0.195 | **0.268** | 0.203 |

pared to NerfStudio (Tancik et al., 2023), 3DGS+MI and our method naturally outperforms ThermoNeRF. Our three thermal Gaussian methods outperform 3DGS+MI across all scenes in PSNR, SSIM, and LPIPS. Among them, ours$_{OMMG}$ shows an average PSNR improvement of 1.3 dB. As shown in Fig. 5, our method's qualitative rendering of thermal images is clearly superior. Additionally, as depicted in Fig. 5f, we enhance thermal image readability by training with MSX images using thermal Gaussian. This hierarchical and easily recognizable thermal Gaussian further promotes the application of thermal scene reconstruction. We provide more qualitative comparison results in the supplementary materials.

## 5.3 RGB VIEW SYNTHESIS

Our method not only achieves high-quality thermal image rendering but also significantly enhances RGB image rendering quality. As shown quantitatively in Table 4, our multimodal constraints improve RGB rendering quality in nearly all scenarios, with an average PSNR improvement of 1.1 dB compared to the original 3DGS. This improvement is particularly evident in scenarios where the RGB modality struggles to identify the environment, while the thermal modality can recognize it clearly. As shown in the top of Fig.6, where distinguishing between foreground and background is challenging in the RGB modality but straightforward in the thermal modality due to temperature differences, constraints from the thermal modality aid in the accurate learning of the RGB modality. Additionally, as depicted in the bottom of Fig.6, the assistance from thermal images enables accurate color rendering in low-light scenes for the RGB modality. Our results demonstrate that, under multimodal constraints, when one modality fails, our approach leverages accurate information from the other modality to enhance the model's understanding of the scene, thus facilitating the correct learning of the failing modality. This enables our method to advance 3D reconstruction in low-light scenes and enhances the robustness of 3D reconstruction techniques to some extent.

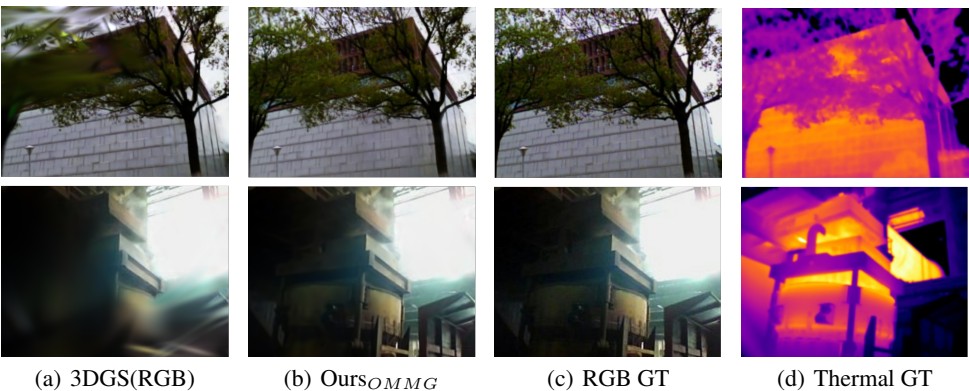

| (a) 3DGS(RGB) | (b) Ours$_{OMMG}$ | (c) RGB GT | (d) Thermal GT |

Figure 6: We present qualitative RGB image comparisons between our method and 3DGS.

Table 4: **Ablation Study**. We conducted ablation experiments by gradually adding each component to the baseline 3DGS model. We then performed a comprehensive comparison across various dimensions, including rendering capability, the quality of rendered color and thermal images, training time, model memory usage, and the number of Gaussians. "-" indicates that the model lacks the specified capability or metric, "×" denotes a reconstruction failure. Since multimodal regularization relies on the Gaussians from multiple modalities, it only applies to Ours$_{MSMG}$.

| Methods | Mode | Thermal | | | RGB | | | Train | FPS | Mem. |
|---|---|---|---|---|---|---|---|---|---|---|
| | T / RGB | PSNR | SSIM | LPIPS | PSNR | SSIM | LPIPS | | | |
| 3DGS | ✓ / - | × | × | × | - | - | - | × | × | × |
| | - / ✓ | - | - | - | 23.20 | 0.820 | 0.219 | 507s | 231 | 159MB |
| +MI | ✓ / - | 24.31 | 0.859 | 0.206 | - | - | - | 367s | 277 | 65MB |
| +MI+$\mathcal{L}_{smooth}$ | ✓ / - | 24.65 | 0.867 | 0.198 | - | - | - | 603s | 292 | 61MB |
| Ours$_{MFTG}$ | ✓ / - | 24.70 | 0.871 | 0.191 | - | - | - | **491s** | 316 | 51MB |
| Ours$_{MSMG}$ | ✓ / ✓ | 25.38 | **0.883** | 0.180 | 24.27 | **0.845** | 0.204 | 873s | 330 / 298 | 18MB+66MB |
| Ours$_{OMMG}$ | ✓ / ✓ | **25.64** | **0.883** | **0.169** | **24.34** | 0.800 | **0.203** | 838s | 271 / 242 | 136MB |
| Ours$_{MSMG}$+MR | ✓ / ✓ | 25.09 | 0.880 | 0.189 | 24.21 | 0.840 | 0.235 | 760s | **390 / 420** | **9MB+9MB** |

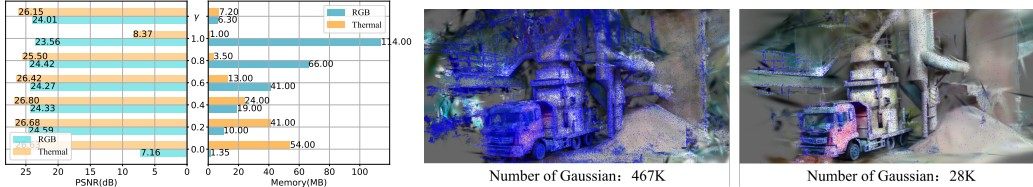

(a) MR ($\gamma$) vs. Fixed coefficient     (b) Gaussian distributions. Left: 3DGS; Right: Ours$_{MSMG}$+MR

Figure 7: Effectiveness of the multimodal regularization term.

## 5.4 ABLATION STUDY

We separate different contributions and algorithm choices to test their effectiveness. As shown in Table 2 and Table 4, after incorporating multimodal initialization, allows 3DGS to achieve thermal reconstruction across various environments. Our multimodal thermal Gaussian models, MSMG and OMMG, not only render both thermal and RGB images simultaneously but also improve rendering quality for both modalities in all scenes, with an average increase of over 1.2 dB. We also observed that multimodal constraints mitigate the generation of excessive redundant Gaussians. Later, we introduced a regularization term to dynamically adjust the coefficients of both modalities. As shown in Table 4, directly training RGB modality Gaussians with 3DGS results in an average storage requirement of 159 MB. On the other hand, directly training thermal Gaussians with MI requires an average of 65 MB. The RGB Gaussians for MSMG+MR average only 9 MB in storage, with thermal Gaussians averaging the same. Our method requires only 8% ($\frac{9+9}{159+65} = 0.08$) of the storage space compared to directly using 3DGS. Due to the reduction in the number of Gaussians, the rendering speed has also significantly increased. Additionally, the rendering quality for both modalities has also improved. MFTG, MSMG+MR, and OMMG excel in different aspects: training speed, storage efficiency, and rendering quality. In Fig.7 a, we compare our multimodal regularization $\gamma$ with manually adjusting the thermal constraint coefficients in a truck scene. The comparison shows that our multimodal regularization approach reduces storage space for both RGB and thermal modalities while maintaining high image quality. In Fig.7 b, we visually present the Gaussian distributions of the original 3DGS method and our method with multimodal regularization.

## 6 CONCLUSION AND FUTURE WORK

We are the first to implement thermal reconstruction based on 3DGS. We not only achieve simultaneous rendering of thermal and RGB images but also significantly improve the rendering quality of both color and thermal images. Additionally, we greatly reduce the model's storage requirements. In the appendix, we discuss the limitations of this work and potential directions for future research.

ACKNOWLEDGMENTS

This work was supported by the Key Project of National Nature Science Foundation of China (U22A2047), the "Pioneer" and "Leading Goose" R&D Program of Zhejiang Province(2023C01044, 2024C01107, 2023C01030, 2023C03012), the National Nature Science Foundation of China (62301198, 62371173).

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
