# OpenReview forum: "ThermalGaussian: Thermal 3D Gaussian Splatting"
_ICLR.cc/2025/Conference — ICLR 2025 Poster_

### Official Review · Reviewer_4Hay · 2024-10-24

**Soundness:** 3
**Presentation:** 3
**Contribution:** 2
**Rating:** 6
**Confidence:** 4

**Summary:**

This paper studies the problem of scene reconstruction with both RGB and thermal image inputs. Specifically, 3D Gaussian Splatting is adopted as the baseline method as a replacement to NeRF. To equip 3D Gaussian Splatting with the ability to also learn 3D thermal representations, this paper proposes a series of strategies for multimodal Gaussian reconstruction, including multimodal initialization, three different thermal Gaussians, constraints specific to thermal modalities, and multimodal regularization. The paper also introduces a new dataset designed for thermal 3D reconstruction.

Experiments demonstrate the advantage of the proposed method in the quality of reconstruction, in both RGB and thermal modalities.

**Strengths:**

The paper investigates an interesting problem of incorporating thermal information into 3D reconstruction by improving on the prevalent 3D Gaussian Splatting method. The presentation is clear and easy to follow. The results are also interesting in that both RGB and thermal reconstruction have improved.

**Weaknesses:**

I do not hold critical concerns to this paper. However, there are some issues that I'd be happy to see clarified.

In the Multiple Single-Modal Gaussians strategy, it is stated that "Each Gaussian model is influenced
not only by its corresponding input modality but also by others." But how so? RGB modality and thermal modality seem to be uncorrelated to me since 3D Gaussians are trained separately for each.

Regarding experiments, is there a specific reason that Thermal-NeRF has not been included in the comparison?

**Questions:**

Please respond to the points made in the Weaknesses section.

---

> ### Author Response · Authors · 2024-11-25
> **Response to Reviewer  4Hay**
>
> We are truly grateful for your feedback. Below are our responses to your comments.
>
> **Q1: RGB modality and thermal modality seem to be uncorrelated to me since 3D Gaussians are trained separately for each.**
>
> A1: The strategy of using Multiple Single Modal Gaussians entails creating two distinct Gaussian distributions for the various modalities. When calculating the loss, the losses from both modalities are weighted and summed using a multimodal regularization strategy. This summed loss is then propagated back into both Gaussians.
>
> We believe that the reconstruction of thermal modality is influenced not only by the input from the thermal modality but also by the input from the color modality. The reconstruction of the color modality is similarly influenced by the thermal modality. Our method's mutual influence enables effective performance even in challenging conditions, such as low light or when distinguishing between foreground and background is difficult, as demonstrated in Fig. 6 of the paper.
>
>
> **Q2: Is there a specific reason that Thermal-NeRF has not been included in the comparison?**
>
> A2: Thermal-NeRF has not been open-sourced yet. Moreover, their dataset only contains the thermal modality without the corresponding color modality. Consequently, our method cannot be tested with their dataset. As a result, we have not performed a direct comparison with Thermal-NeRF.
> Our project will be open-sourced to encourage comparisons and promote further advancements in this field.

---

> ### Comment · Reviewer_4Hay · 2024-11-27
>
> Thanks for the authors' response, it addressed most of my concerns. The reviewer will keep the original rating.

---

> > ### Author Response · Authors · 2024-11-27
> > **Thank you for maintaining the score.**
> >
> > Thank you for recognizing the contributions of our work and maintaining your original score. We sincerely appreciate your clear and positive feedback, particularly your remark, *"I do not hold critical concerns to this paper."*

---

### Official Review · Reviewer_5amv · 2024-11-02

**Soundness:** 3
**Presentation:** 2
**Contribution:** 2
**Rating:** 6
**Confidence:** 4

**Summary:**

ThermalGaussian is the first work with thermal 3D reconstruction with 3DGS representation.

The paper achieves thermal and RGB 3D reconstruction simultaneously, and improves the rendering results both of color and thermal images.

Finally, the paper reduces the model storage cost by 90%.

**Strengths:**

The paper presents new thermal-RGB datasets which will be helpful in future research.

The method shows quality improvement not only for thermal reconstruction but also for RGB reconstruction.

The method that adds thermal reconstruction on 3DGs is effective and straightforward.

**Weaknesses:**

The paper proposes three multimodal training methods, which are MFTG, MSMG, and OMMG. The reviewer is confused about the pros and cons of each model, and users are confused about which model to choose. In the quantitative evaluation, the best performing models are distributed across each scene, with no model performing overwhelmingly well.

In the three multimodal training methods, the insight about why these three design are needed is lacked. In L. 283, the authors say “thermal Gaussian training in the second phase may not fully leverage the information from the color modality”, and there is no evidence about this.

There is no discussion about the limitations.

In L.448, there is a type. “Ours” → “ours”

**Questions:**

In the paper, the authors say that thermal imaging is practical task and 3D reconstruction gives significant applications which 2D images cannot provide. The reviewer wonders what is the specific case of that thermal 3D reconstruction is practical than 2D thermal image? There is a logical jump in the connection between the two. This context would be added at the related work 2.1 section.

In L. 314, the paper denotes thermal equilibrium, and it would be nice to explain what this means.

In the related work section, both of ThermoNeRF and ThermalNeRF are denoted but why only ThermoNeRF is compared to your method?

The reviewer suggests that comparison to existing thermal 3D reconstructions and to datasets used in previous studies, such as ThermoNeRF and Thermal-NeRF, would further emphasize the strength of the method itself.

---

> ### Author Response · Authors · 2024-11-25
> **Response to Reviewer 5amv #1**
>
> We sincerely thank the reviewer for the positive feedback and valuable suggestions.  Here are our responses to your comments.
>
>
> **Q1: The paper proposes three multimodal training methods, which are MFTG, MSMG, and OMMG.**
>
> A1: We apologize for the clarification problem. In the revised version, we have highlighted the distinct benefits of each of the three multimodal strategies in **Sec. 5.4**. As shown in **Tab. 4**, different methods exhibit distinct strengths:
>
> **1. MFTG:** When a high-quality geometric point cloud of the scene is available, MFTG enables faster reconstruction of the temperature field, reducing training time by 40\% compared to the other methods.
>
> **2. MSMG with Multimodal Regularization:** This approach greatly minimizes storage needs for both modalities during reconstruction, making it more efficient for resource-constrained applications.
>
> **3. OMMG:** In most cases, OMMG provides the highest rendering quality for both color and temperature modalities. We hope these clarifications address your concerns and offer a clearer understanding of the strengths of each approach.
>
> **Q2: In the three multimodal training methods, the insight about why these three designs are needed is lacking.**
>
> A2: We sincerely thank the reviewer for the reminder. In Sec. 5.4, we elaborated on the advantages of three distinct strategies introduced in our work, as they address different application needs. In our response to A1, we clarified the strengths of each multimodal training strategy: **MFTG:** Offers faster training times. **MSMG:** Reduces storage requirements.
> **OMMG:** Delivers superior novel-view rendering quality in most scenarios.
>
> We apologize for any confusion caused by the statement in L. 283. The sentence outlines the MFTG strategy, which consists of a two-stage approach: the first stage uses only the RGB modality, and the second stage relies solely on the thermal modality. In contrast, both MSMG and OMMG simultaneously use RGB and thermal data. Therefore, when we stated that "MFTG does not fully utilize the RGB modality," we were specifically referring to its second stage, where RGB data is not used for supervision.
>
> **Q3: There is no discussion about the limitations.**
>
> A3:  Thank you for the insightful suggestion. Our work builds on the 3DGS framework to accomplish thermal and RGB reconstruction. Like many methods based on 3DGS, it encounters major challenges in dynamic scenes, reflective environments, and low-texture scenarios. Due to the limitations of infrared thermal imaging, which only measures the temperature of solid surfaces, our method is currently restricted to reconstructing and measuring surface temperatures instead of estimating temperatures at arbitrary points in space. We have added the discussion of limitations to **Appendix B**.
>
> **Q4: In L.448, there is a type. “Ours” → “ours”**
>
> A4: We sincerely thank the reviewer for pointing out this issue. In the revised version, we have corrected this error and have conducted a thorough review of the entire manuscript to ensure that similar issues do not occur elsewhere.
>
> **Q5: The reviewer wonders what is the specific case of that thermal 3D reconstruction is practical than 2D thermal image?**
>
> A5: Thermal 3D reconstruction is more practical than 2D thermal imaging in at least two major types of scenarios:
>
> 1. Scenarios That Require Detailed Temperature Analysis: Multiple 2D thermal images taken from different angles can capture an object's temperature distribution. By reconstructing these images into a 3D model, observers can examine the temperature distribution of the object from any perspective rather than being limited to the predefined angles at which the images were captured. This offers a clearer and more intuitive representation of the object, facilitating more accurate and comprehensive thermal analysis.
>
> 2. Large-Scale Scenario Analysis: In large environments, a single 2D image cannot fully capture the entire scene. Even with several images, observers find it difficult to understand the temperature distribution across the entire area.
>
> Thermal 3D reconstruction greatly surpasses standalone thermal images or videos, offering deeper insights and facilitating more accurate, informed decision-making.
> In the revised version, we will enhance the description of 3D reconstruction versus 2D imaging in **Sec. 2.1** to eliminate logical gaps and we have provided examples of thermal 3D reconstruction applications to help readers better understand the potential use cases of our work. These examples will be included in the **Appendix C**.

---

> > ### Author Response · Authors · 2024-11-25
> > **Response to Reviewer 5amv #2**
> >
> > **Q6: In L. 314, the paper denotes thermal equilibrium, and it would be nice to explain what this means.**
> >
> >
> > A6: We sincerely thank the reviewer for this valuable suggestion.
> > Thermal equilibrium occurs when two systems stop the transfer of heat from the hotter system to the cooler one, resulting in both achieving equal temperatures.
> > In the revised version, we have added an explanation of the term thermal equilibrium at its first occurrence in **Sec. 3.2** to improve clarity for the readers.
> >
> > **Q7: In the related work section, both of ThermoNeRF and Thermal-NeRF are denoted but why only ThermoNeRF is compared to your method?**
> >
> > A7: Thermal-NeRF has not been open-sourced yet. Moreover, their dataset only contains the thermal modality without the corresponding color modality. Consequently, our method cannot be tested with their dataset. As a result, we have not performed a direct comparison with Thermal-NeRF.
> > Our project will be open-sourced to encourage comparisons and promote further advancements in this field.
> >
> >
> > **Q8: The reviewer suggests that comparison to existing thermal 3D reconstructions and to datasets used in previous studies.**
> >
> > A8: The ThermoNeRF dataset does not have fixed temperature limits during data capture, leading to temperature variations when observing the same object from different angles. This results in insufficient multi-view consistency, failing the 3D reconstruction requirements.
> > The Thermal-NeRF dataset contains only thermal data and does not include any RGB data.
> > Our dataset includes consistent multi-view data and pre-registered color and thermal images, along with camera poses obtained from COLMAP for multi-modal reconstruction.

---

> > > ### Comment · Reviewer_5amv · 2024-11-27
> > >
> > > Thank you for the authors' responses. The reviewer decided to raise the score because the authors addressed my concerns well. The reviewer appreciates the detailed explanations of the authors.

---

> > > > ### Author Response · Authors · 2024-11-27
> > > > **Thank you for reviewing our response and raising the score.**
> > > >
> > > > Thank you very much for recognizing our work and raising the score. We are really delighted that your valuable feedback greatly helps improve the completeness of our work.

---

### Official Review · Reviewer_RJXA · 2024-11-03

**Soundness:** 3
**Presentation:** 3
**Contribution:** 3
**Rating:** 8
**Confidence:** 3

**Summary:**

This paper proposes an enhanced method that extends the 3D Gaussian Splatting (3DGS) technique through the ThermalGaussian model, allowing for joint learning and rendering of RGB and thermal data. By learning thermal images together with RGB data in a multimodal framework, the approach aims to improve 3D scene reconstruction capabilities across fields where thermal imaging is valuable, such as in military, industrial, and medical applications. The paper presents initialization strategies for aligning and merging RGB and thermal data, a loss function tailored to the characteristics of thermal imaging, and multimodal regularization techniques to prevent overfitting and optimize storage. Additionally, a new dataset, RGBT-Scenes, is introduced, providing research data that includes both RGB and thermal images.

**Strengths:**

1. Effective Multimodal Learning and Loss Function Design

The paper introduces an initialization strategy for aligning RGB and thermal data, along with a loss function tailored to thermal characteristics, allowing these modalities to learn complementarily. This approach enables each modality to mitigate the limitations of the other, achieving optimal reconstruction performance​.

2. Model Efficiency and Storage Optimization

By applying multimodal regularization to prevent overfitting and reduce redundant Gaussians, the model reduces storage requirements by approximately 90% compared to conventional methods. This enables high-quality 3D reconstruction while also enhancing storage efficiency and rendering speed​.

2. Contribution of the RGBT-Scenes Dataset to the Research Community

The RGBT-Scenes dataset introduced in this paper provides paired RGB and thermal images across diverse indoor and outdoor scenes, offering a valuable resource for thermal-based 3D reconstruction research. This dataset provides a foundation for researchers to test and expand upon the proposed model, thereby promoting advancements in the field​.

**Weaknesses:**

1. Lack of Thermal Data Description

While the provision of a dataset is one of the significant contributions of this paper, it lacks sufficient information for readers to understand the characteristics of thermal data. Including details about general characteristics of infrared imaging, as well as optical or photographic settings used during data collection (e.g., imaging wavelength range, whether exposure values were fixed), would help readers better understand the dataset's properties and enhance reproducibility.

2. Lack of Discussion on Limitations

The paper focuses on the model's performance but lacks an explicit discussion of its limitations. For example, it would benefit from exploring conditions under which the proposed multimodal regularization and initialization strategies may not perform well. Including such a discussion would allow for a more realistic assessment of the model's applicability and provide important insights for researchers who may wish to build upon this work.

3. Lack of Discussion on Applicability and Future Work
Although the multimodal learning approach for RGBT data proposed in this paper is academically valuable, the discussion on future work is somewhat general. The paper mentions super-resolution and dynamic scene reconstruction as directions for further research, but these are relatively standard research topics rather than areas specifically tailored to advancements in thermal 3D reconstruction.

**Questions:**

1. Applicability to a Broader Range of Hyperspectral Imaging

While the RGBT images presented involve the addition of a single channel, they can broadly be considered a form of hyperspectral imaging, as they incorporate the infrared spectrum. Could the authors discuss the potential for their multimodal approach to be applied to a broader range of hyperspectral imaging? Additionally, I am interested in the authors' views on whether this approach could be extended to studies involving multi-wavelength, multichannel data, and if it could serve as a basis for hyperspectral data analysis.

2. Requirement for Well-Aligned RGB and Thermal Data and Practical Field Applicability

The proposed method relies on the availability of well-aligned RGB and thermal data, which may limit practical field applications. In cases where precise data alignment is challenging or unfeasible in real-world scenarios, are there any methods the authors could suggest to mitigate this limitation?

**Details Of Ethics Concerns:**

It is necessary to ensure that the dataset was acquired properly.

---

> ### Author Response · Authors · 2024-11-25
> **Response to Reviewer RJXA #1**
>
> We sincerely appreciate the reviewer's positive feedback and valuable suggestions.
>
> **Q1: Lack of Thermal Data Description.**
>
> A1:  Thank you for the suggestion. While **Sec. 2.1** of the original manuscript briefly introduced the principles and applications of thermal imaging, it lacked specific descriptions of thermal images. In the revised version, we have enriched **Sec. 2.1** by adding fundamental information about thermal imaging, including common color rendering settings and typical imaging wavelengths. Furthermore, to assist readers in collecting similar datasets, we have included detailed descriptions of the cameras and their settings used during data collection in the  **Appendix A**. Additionally, the datasets we have made available include the raw data collected, which will help future researchers reference and utilize them.
>
> **Q2: Lack of Discussion on Limitations.**
>
> A2:  Thank you for the insightful suggestion.  Our work builds on the 3DGS framework to accomplish thermal and RGB reconstruction. Like many methods based on 3DGS, it encounters major challenges in dynamic scenes, reflective environments, and low-texture scenarios. Due to the limitations of infrared thermal imaging, which only measures the temperature of solid surfaces, our method is currently restricted to reconstructing and measuring surface temperatures instead of estimating temperatures at arbitrary points in space. We have added the discussion of limitations to **Appendix B**.
>
> **Q3: Lack of Discussion on Applicability and Future Work.**
>
> A3: Thank you for your insightful suggestion.  We have added the discussion of applicability and future work to **Appendix C** and **Appendix D**. The hyperspectral 3D reconstruction mentioned in Q4 represents a valuable direction for future research.  As for RGB-thermal fields, we added a more detailed discussion of applications and future work:
>
> **Applications of RGB-thermal fields:** In Sec. 2.1, we briefly discussed various applications of thermal imaging. We believe that any application utilizing thermal imaging can enhance the observation and analysis of temperature distributions by establishing RGB-thermal fields. For example: 1. Building energy analysis: Reconstructing a building's 3D thermal field aids in evaluating energy efficiency. 2. Battlefield analysis: The integration of 3D thermal and RGB reconstructions of specific battlefield scenarios enhances strategic planning. 3. Equipment monitoring: Improving fault prevention and diagnosis can be achieved by reconstructing 3D thermal fields for high-voltage power equipment and high-temperature devices. 4. Fire rescue: Reconstructing the 3D thermal field of fire scenes assists rescue teams in devising optimal strategies to save lives.
>
> **Future works of RGB-thermal fields:** At present, our method is restricted to reconstructing surface temperatures. In future work, we plan to investigate reconstruction beyond surface thermal fields, including events such as flames. High-resolution thermal imaging devices tend to be expensive, while high-resolution RGB cameras are generally more affordable. We will explore methods to use high-resolution RGB images to enhance the super-resolution of low-resolution thermal images. This approach aims to provide high-resolution thermal imaging while significantly reducing costs. Many thermal scenes are dynamic. To improve the reconstruction of such scenes, we plan to extend our work to include dynamic field reconstruction.
>
> **Q4: Applicability to a Broader Range of Hyperspectral Imaging.**
>
> A4: Thank you for highlighting the importance of hyperspectral imaging.  We believe our method can be applied to hyperspectral imaging and possibly any imaging modality, provided the camera pose estimation challenge is addressed.
> For images with rich textures, existing tools like COLMAP can effectively estimate camera poses. However, for images with low textures, our method utilizes multimodal initialization by leveraging the high-texture RGB modality. After our review, we found that hyperspectral images generally have relatively low texture.
> Hyperspectral imaging typically involves materials that reflect or absorb specific spectral information, providing opportunities for calibration.
> By applying materials that reflect specific spectral bands to the white regions and absorb them in the black regions of a checkerboard pattern, it is possible to create a calibration board for hyperspectral cameras. Using our multimodal initialization method, the hyperspectral modality can be aligned with the RGB modality to estimate camera poses. Subsequently, our approach can be applied to hyperspectral imaging reconstruction.

---

> > ### Author Response · Authors · 2024-11-25
> > **Response to Reviewer RJXA #2**
> >
> > **Q5: Requirement for Well-Aligned RGB and Thermal Data and Practical Field Applicability.**
> >
> > A5: Our approach uses aligned texture-rich RGB images and thermal images to compute the camera poses and intrinsic parameters for the thermal camera. If the RGB and thermal images are misaligned, we recommend using only the thermal images and applying low-texture-compatible COLMAP [1] to estimate the camera poses and intrinsic parameters.
> >
> > [1] He X, Sun J, Wang Y, et al. Detector-free structure from motion[C]//Proceedings of the IEEE/CVF Conference on Computer Vision and Pattern Recognition. 2024: 21594-21603.

---

> > > ### Comment · Reviewer_RJXA · 2024-12-02
> > >
> > > Dear Author,
> > >
> > > I believe that the multi-modality methodology proposed by you is effective for Thermal-RGB image pairs and that the experiments conducted to verify this are appropriate. Therefore, I stand by my evaluation.

---

> ### Author Response · Authors · 2024-12-02
> **Thank you for maintaining the score.**
>
> We appreciate your recognition of our work and thank you for the key insights that further enhanced the quality of the paper. Additionally, your suggestion to explore 3D reconstruction using hyperspectral imaging has introduced an exciting new research direction.

---

### Official Review · Reviewer_teJM · 2024-11-03

**Soundness:** 3
**Presentation:** 3
**Contribution:** 2
**Rating:** 5
**Confidence:** 4

**Summary:**

This paper proposes a method based on 3DGS that jointly learns multi-modal scene representations of RGB and thermal images. In fact, the authors present and evaluate three different strategies to incorporate thermal images into 3DGS: MFTG (based on fine-tuning), MSMG (leverages multi-task learning), and OMMG—the latter yields a single Gaussian, which is achieved by extending 3DGS with spherical harmonics to represent thermal data. Moreover, a regularization term is proposed that accounts for the smooth nature of thermal images, and a new weight scheduling strategy prevents the proposed MSMG from overfitting to a single modality. Based on a newly collected RGB+thermal dataset consisting of ten scenes, the authors could successfully show that their method outperforms 3DGS baselines and a recent NeRF-based method in terms of RGB and thermal rendering quality while requiring less memory.

**Strengths:**

- The topic is interesting and of importance to the community, as demonstrated by the large body of recent work [1-5].

- The paper is well-written and easy to follow. Experiments are well described and cared out thoroughly. The ablation study nicely supports all design decisions. All in all,  I don’t see any obvious flaws.

- This is (to the best of my knowledge) the first paper to use some kind of scheduling to dynamically adjust the weight that balances the RGB and thermal loss in Eq. (12). The proposed strategy seems plausible to me, and, according to the experimental evaluation, significantly reduces the memory footprint while only slightly decreasing image quality. Due to the reduced number of Gaussians, this strategy also increases rendering speed.

- Compared to prior work [1,3] (but similar to [5]), the method presented in this work is actually capable of *improving* the RGB rendering quality, especially in low-light conditions.

- The paper proposes and evaluates not only a single method but three different strategies to incorporate thermal images into 3DGS, thus can be seen as the 3DGS-based pendant to [1].

- I appreciate the newly collected dataset. Although small-scaled (like all other publicly available RGB+thermal datasets), it is diverse, contains forward-facing and 360-degree scenes, and encompasses indoor and outdoor environments. This is very helpful to the community.

References:

[1] Mert Özer et al. Exploring Multi-modal Neural Scene Representations With Applications on Thermal Imaging. ECCVW, 2024.

[2] Yvette Y. Lin et al. ThermalNeRF: Thermal Radiance Fields. ICCP, 2024.

[3] Miriam Hassan et al. ThermoNeRF: Multimodal Neural Radiance Fields for Thermal Novel View Synthesis. arXiv, 2024.

[4] Tianxiang Ye et al. Thermal-NeRF: Neural Radiance Fields from an Infrared Camera. arXiv, 2024.

[5] Jiacong Xu et al. Leveraging Thermal Modality to Enhance Reconstruction in Low-Light Conditions. ECCV, 2024.

**Weaknesses:**

- I am missing citations (and discussions) of recent related works [1,2,5] in the introduction and related work sections.

- I would love to see empirical evidence for the statement in L337: "we observe that the weight coefficients of a modality align linearly with the Gaussian it ultimately generates". This is a quite interesting finding that would be much better supported with data.

- Authors claim multiple times (for example in L028, L104, and L536) that they are the first to use 3DGS to model RGB and thermal images. This statement is actually not true and should be revised. The authors of [5] already presented a 3DGS-based method that incorporates RGB and thermal data (see Section 4.5 of the main paper and Section 4 of supplementary).

- In Table 1, the following open-source multi-view RGB+thermal datasets are missing: ThermalMix proposed in [1], the datasets presented in [2], and MVTV proposed in [5]. Why is the proposed dataset better than these datasets? Also, it would be good to provide specific criteria for how multi-view consistency and content richness is evaluated across datasets. This would help clarify the comparison and justify any claims about the proposed dataset's advantages.

- With the high number of recently published RGB+thermal datasets, I think it might make sense to evaluate the introduced method on at least one of these datasets, also to mitigate potential dataset bias.

- In addition, the proposed method could be compared to some more of the recently published NeRF-based methods. There is code available for [2].

- In L097 authors mention that ThermoNeRF [3] reduces RGB rendering quality, while the proposed method improves it (according to Table 3, an average increase of almost 5% in PSNR). I think that is a huge plus for the proposed method; however, this argument would be much stronger if authors would add comparisons with NeRF-based methods to Table 3.

- I would love to see a side-by-side comparison of thermal renderings with and without the smoothness term.

- Although the proposed weight scheduling used in MSMG yields lower storage requirements, the average rendering quality for RGB and thermal data is worse than for OMMG (see Tables 2 and 3; difference higher for thermal renderings, drop of about 1% in PSNR). So, if one wants to maximize rendering quality and therefore uses OMMG, the benefit of having lower memory requirements is gone (see Table 4; OMMG requires about 60% of the storage space of 3DGS in contrast to 8% for MSMG).

- In Section 3.2, three different strategies to obtain camera poses for thermal images are mentioned. What strategy is actually used for the experiments? Also, what strategy performs best? A quantitative comparison would be really helpful, also because MSX seems to perform best (just by looking at Figure 2). The problem I see is that MSX is patented by FLIR, as such it is not available to other cameras. Hence, the practical use of the third strategy is highly limited, and it would be good to know which strategy could be used instead. Do the other strategies in any form make use of MSX images as well?

References:

See above.

**Questions:**

- What does „Mix50“ in Figure 2 stand for? Is Eq. (5) applied with $\beta=0.5$? If so, please leave a note in the caption of Figure 2.

- How sensitive is $\lambda_{smooth}$? Would it be possible to provide a small ablation study?

- How was $\lambda$ in Eq. (6) and Eq. (10) set? Also, are these two lambdas the same?

- As far as I understand, MSMG uses the cost function described in Eq. (7), i.e., without multimodal regularization. So, whenever "Ours_MSMG" is used in Tables 2 and 3, it refers to MSMG method trained without multimodal regularization. This is in accordance with Table 4, where authors add a dedicated row for "Ours_MSMG+MR" which would not be necessary if MSMG uses MR per default (in this case, I would rather expect a row with "Ours_MSMG w\o MR" in Table 4). As this would mean that MR has never been used except in the ablation study, I would guess that per default, Eq. (12) has been used to train MSMG. Please make this more clear (e.g., by replacing Eq. (7) with Eq. (12) and referring to Section 3.4. on how to choose $\gamma$).

- What cost function is used to train OMMG? Is it Eq. (7)? If so, please move it to the respective paragraph (starting L296).

- In Figures 5 and 6, which strategy does "Ours" refer to?

- To be honest, it is quite surprising to me that a conventional chessboard pattern printed on normal paper produces such highly contrasting thermal images. There are numerous works (see, e.g., [6] and references therein) that try to design special calibration objects for thermal cameras. I wonder why all these papers exist when a simple chessboard pattern printed on paper will also work. What are the physical principles behind why a simple chessboard pattern printed on paper works, and how does it compare to more complex calibration objects in terms of accuracy and ease of use?

References:

[6] Issac N. Swamidoss et al. Systematic approach for thermal imaging camera calibration for machine vision applications. Optik 247, 2021.

---

> ### Author Response · Authors · 2024-11-25
> **Response to Reviewer  teJM #1**
>
> We sincerely appreciate the reviewer's positive feedback and valuable suggestions. Below are our responses to your comments.
>
> **Q1: I am missing citations (and discussions) of recent related works [1,2,5] in the introduction and related work sections.**
>
> A1: We also value the important contributions of the related works you referenced. In the revised version of our paper, we have cited these works [1, 2, 5].  Here, we outline the differences between our work and the three papers mentioned.
>
> [1] is a notable NeRF-based multi-modal reconstruction project that, similar to ours, introduces various multi-modal strategies. Their strategies often reduce the reconstruction quality of the RGB modality, whereas our multi-modal approaches improve the reconstruction results for both modalities simultaneously. Our multi-modal regularization effectively reduces overfitting in reconstruction, decreases storage needs, and enhances both training and rendering speed.
>
> [2] is a successful work on NeRF-based thermal field reconstruction, showcasing various potential applications of thermal imaging and reconstruction. Like our work, it also focuses on aligning thermal and RGB images. Unlike them, we utilize a standard checkerboard for RGB-thermal alignment, which makes our approach more accessible and easier for broader adoption than their custom-designed calibration board.
>
> [5] focuses on NeRF-based RGB reconstruction under low-light conditions. Like our research, [5] shows that thermal modalities can improve RGB reconstruction quality, especially during low-light conditions. However, it largely overlooks the importance of reconstructing the thermal modality itself. In contrast, our work enhances RGB reconstruction in low-light conditions and also provides a reconstructed thermal field suitable for thermal analysis.
>
> We hope these clarifications emphasize the unique contributions of our work and the complementary value of concurrent studies in advancing this exciting field of research.
>
> **Q2: I would love to see empirical evidence for the statement in L337.**
>
> A2: Thank you for highlighting this intriguing observation.   From the right side of Fig. 7 (a), we see that the storage space required for each modality correlates directly with the corresponding coefficient. As the weight for the color modality increases, the storage space and the number of Gaussian points for color also increase. The same trend is observed for the thermal modality. This observation led us to propose a multi-modal regularization strategy that dynamically balances these weights for optimal results.
>
> **Q3: Authors claim multiple times (for example in L028, L104, and L536) that they are the first to use 3DGS to model RGB and thermal images.**
>
> A3: We compare the difference between [5] and our work in A1.
>
> **Q4:  In Table 1, the following open-source multi-view RGB+thermal datasets are missing...**
>
> A4: Thank you for your reminder. We will incorporate these concurrent works and their datasets in the revised version. Regarding the datasets:
>
> [1]: The open-source dataset provides only grayscale thermal images, unlike the color-mapped thermal images presented in their paper.
>
> [2]: This study is most closely related to ours, providing suitable thermal and color images. In the open-source dataset, the two modalities are not aligned, and a custom calibration board was utilized for this alignment. The calibration results were not provided, making the dataset usable only with a hard-coded implementation. In contrast, our dataset includes not only the original captured images but also precisely aligned RGB and thermal image pairs.
>
> [3]: While this work provides RGB and color-mapped thermal images, the dataset shows significant temperature inconsistencies for the same position across various views of the same scene. This inconsistency significantly affects the accuracy of 3D reconstruction.
>
> [5]: This work focuses solely on improving color modality reconstruction under low-light conditions. Therefore, their dataset provides only grayscale thermal images without the commonly used color-mapped thermal renderings.
>
> **Q5: With the high number of recently published RGB+thermal datasets, I think it might make sense to evaluate the introduced method on at least one of these datasets, also to mitigate potential dataset bias.**
>
>
> A5: We appreciate your suggestion. However, we cannot currently utilize the datasets from the other works you mentioned.
>
> [1, 5]: These datasets contain only grayscale thermal images; they do not include color-mapped thermal images.
>
> [2]: This dataset does not establish a correspondence between the color and thermal modalities.
>
> [3]: The thermal images in this dataset lack consistency across views, as the temperature for the same position varies.
>
> [4]: This dataset includes only thermal images without corresponding color images.
>
> These datasets cannot be directly used for our comparisons or analyses due to their limitations.

---

> > ### Author Response · Authors · 2024-11-25
> > **Response to Reviewer teJM #2**
> >
> > **Q6: In addition, the proposed method could be compared to some more of the recently published NeRF-based methods. There is code available for [2].**
> >
> > A6: The ThermalNeRF [2] team has conducted impressive work on thermal field reconstruction, and we are very interested in comparing our methods with theirs. However, their hard-coded implementation currently limits the code to work exclusively with their own dataset, making it difficult to test on external data. As stated in their documentation: *“This script does currently hard-code some assumptions. Changing the hard-coded nature of these quirks is on our to-do list.”*
> > We reviewed their implementation and identified two specific limitations:
> >
> > 1. Calibration Process: Their code requires running a calibration step involving a custom-made calibration board. This board consists of a 4 x 11 asymmetric grid of circular cutouts, each with a diameter of 15mm and a center-to-center distance of 38mm. Although their code allows for some modification of the parameters for circular calibration boards, it currently does not support the standard square checkerboard calibration boards that we used for our dataset. Modifying their code to accommodate square checkerboards proved challenging due to insufficient documentation.
> >
> > 2. Camera Assumptions: Their code sets the camera spacing between the third and fourth images to exactly 1 meter. This assumption does not align with our dataset, which features images captured with random camera movements to mimic real-world applications. In practical scenarios, images are often captured with freehand motion, making their setup unsuitable for more flexible reconstruction workflows.
> >
> > Furthermore, their open-source dataset lacks pre-aligned color and thermal images, as well as camera pose information obtained using COLMAP. As a result, their dataset is currently only compatible with their hard-coded implementation. This limitation further reduces its usefulness for comparing it with external methods. Given these constraints, we are unable to perform a direct comparison with ThermalNeRF at this stage.
> >
> > **Q7: In L097 authors mention that ThermoNeRF [3] reduces RGB rendering quality...**
> >
> > A7: Thank you for your constructive suggestion, which highlights our contributions. In the revised version, we will include the results from [3] in **Tab. 3**.
> >
> > **Q8: I would love to see a side-by-side comparison of thermal renderings with and without the smoothness term.**
> >
> > A8: Thank you for your reminder. In fact, in **Tab. 4**, "$+\mathcal{L}_{\text {thermal}}$" represents the addition of the smoothness term on top of "$+MI$." Our ablation study was designed to demonstrate a progressive relationship, with each step building on the previous one by introducing new contributions. Tab. 4 presents a comparison of thermal renderings, highlighting the differences between models with and without the smoothness term.
> >
> > **Q9: Although the proposed weight scheduling used in MSMG yields lower storage requirements...**
> >
> > A9: In the current approach, MSMG and OMMG each offer distinct advantages and disadvantages. Users can select the strategy that best meets their individual needs. For large-scale reconstructions of environments like factories, where storage efficiency is critical, MSMG would be the more suitable choice. If higher image rendering quality is needed, the OMMG strategy is preferred.
> >
> > **Q10: In Section 3.2, three different strategies to obtain camera poses for thermal images are mentioned...**
> >
> > A10: We conducted quantitative experiments on various initialization strategies in the **Appendix G**, along with relevant descriptions of these experiments. Our current results show that the best localization performance is achieved using the direct mixing strategy of the two modalities rather than the MSX strategy.
> >
> > |Initialization Modality | Ebike | Iron Ingot | Rotary Kiln | parterre |
> > |:---------:|:---------:|:---------:|:---------:|:---------:|
> > |RGB | 20.89| 29.57| 26.59| 22.09|
> > |Mix($\beta = 0.3$) | 21.09 | 30.15 | 27.11 | 23.02 |
> > |Mix($\beta = 0.5$) | **22.77**| **30.45** | 27.35 | **24.18**|
> > |Mix($\beta = 0.7$) |$\times$ | 12.34|  **27.78**| 22.17|
> > |Thermal | $\times$ | $\times$| $\times$| $\times$ |
> > |MSX | 22.20 | 29.94 |27.30 |24.02|
> >
> > We tested different input images from Sec. 3.2 across several scenes to estimate camera poses. Subsequently, these poses were used in the baseline 3DGS for thermal field reconstruction. The quality of the estimated poses was evaluated by comparing the PSNR values of the reconstructed thermal images from novel views. The experiments demonstrate that mixing the aligned color and thermal modality images with $\beta = 0.5$ produces the most accurate camera poses.

---

> > > ### Author Response · Authors · 2024-11-25
> > > **Response to Reviewer teJM #3**
> > >
> > > **Q11: What does Mix50 in Figure 2 stand for $\beta = 0.5$ Is Eq. (5) applied with? If so, please leave a note in the caption of Figure 2.**
> > >
> > > A11: Thank you for your suggestion. Mix50 refers to $\beta = 0.5$ in Eq. (5). In the revised version, we have clarified this in **Fig. 2**.
> > >
> > > **Q12: How sensitive is $\lambda_(smooth)$? Would it be possible to provide a small ablation study?**
> > >
> > > A12: Similar to our response to A8. In fact, in **Tab. 4**, "+$\\mathcal{L_smooth}$" represents the addition of the smoothness term on top of "+MI." Our ablation study was designed to demonstrate a progressive relationship, where each step builds on the previous one by adding new contributions. Therefore, Tab. 4 provides a side-by-side comparison of thermal renderings with and without the smoothness term.
> > >
> > > **Q13: How was $\lambda$ in Eq. (6) and Eq. (10) set? Also, are these two lambdas the same?**
> > >
> > > A13: Thank you for your question. The $\lambda$ without subscripts share the same meaning and are consistent with the $\lambda$ in the baseline 3DGS method. Since this $\lambda$ is not an innovation of our work but is directly inherited from 3DGS, we clarified this in the revised version under **Sec. 5.1**, which outlines the details of the experimental implementation.
> > >
> > > **Q14: As far as I understand, MSMG uses the cost function described in Eq. (7)...**
> > >
> > > A14: Thank you for your question. Unless otherwise specified, Ours$_{MSMG}$ refers to the version without multimodal regularization. Since multimodal regularization is a key contribution to our work, we chose to highlight it separately. The setup for multimodal regularization is detailed in **Sec. 3.4**, and its effectiveness is demonstrated through **Tab. 4** and **Fig. 7**. Relevant descriptions are provided in **Sec. 5.4**.
> > >
> > > **Q15: What cost function is used to train OMMG? Is it Eq. (7)? If so, please move it to the respective paragraph (starting L296).**
> > >
> > > A15: Thank you for pointing this out. It is Eq. (7). Since MSMG also uses Eq. (7), we cannot move it elsewhere. We appreciate your suggestion for improving the paper's consistency and readability. We will clarify these points in the revised version.
> > >
> > >
> > > **Q16: In Figures 5 and 6, which strategy does "Ours" refer to?**
> > >
> > > A16: In Figures 5 and 6, the term "Ours" specifically refers to OMMG.  To avoid any potential confusion for readers, we have updated the labels in the revised version to "Ours$_{OMMG}$."  We appreciate your attention to this detail.
> > >
> > > **Q17: To be honest, it is quite surprising to me that a conventional chessboard pattern...**
> > >
> > > A17: Thank you for this intriguing question.  It is important to clarify that the chessboard pattern used is printed on a calibration board made from aluminum oxide material, which is widely available, non-reflective, inexpensive, and commonly used for calibration purposes. During our calibration process, we achieved a mean reprojection error of less than 0.5 pixels, which we consider to be sufficiently accurate for practical applications. The observed phenomenon is probably based on the material's physical properties: During rapid infrared heating, the black regions absorb heat faster than the white regions, resulting in a temporary temperature difference before thermal equilibrium is achieved. This thermal disparity lasts for a few minutes, allowing us to capture thermal images, as shown in Figure 4, for alignment purposes. We consider this finding to be valuable enough to share with the community, as it provides a practical alternative for calibration in multimodal imaging workflows.

---

> ### Comment · Reviewer_teJM · 2024-11-28
>
> I appreciate the author's detailed rebuttal. While some of my concerns have been resolved, the following remain.
>
> A1: Thank you for the clarification. However, I still can’t find this information in the revised version of the paper. I think the author’s response in the rebuttal is nice and should be worked into the paper.
>
> A2: Thanks for pointing this out—I see the trend on the right side of Figure 7(a). I think this evidence nicely supports the claim, but I wish the authors had referred to it earlier, possibly in Section 3.4 when they introduce their multimodal regularization strategy. I still think this is a very nice observation that deserves more attention.
>
> A3: Unfortunately, I can’t find Section A1 in the revised version of the supplementary material.
>
> A4: While I do believe that the color-coding of thermal images in papers [1,5] is only added for visualization purposes after inference, I do have to agree that those datasets may not be used at this point since we don’t know for sure. This is actually quite unfortunate, but certainly not the author's fault. I appreciate the author's clarification. However, I don’t see the point of not using the dataset proposed in [3]. If it is of bad quality, then all methods evaluated on this dataset simply perform equally badly, but the comparison *across methods* would still be fair according to my understanding. Finally, I’m still missing these datasets in Table 1 of the revised paper.
>
> A8: I was actually referring to a side-by-side comparison of thermal *renderings* (i.e., images) with and without the smoothness term. I apologize in case I wasn’t clear enough.
>
> A17: Thanks for the clarification. As this is very important information, I’m wondering why the authors did not include it in the revised version of the paper. To me, it becomes not clear from Section 3.2 that the checkerboard pattern is actually printed on aluminum oxide material. Is there a special printer required to print the checkerboard pattern onto the aluminum oxide? Honestly, I still don’t see the benefit of the proposed calibration device compared to a simple perforated aluminum plate (which is also widely available and inexpensive). I would appreciate further clarification on this point.
>
> Overall, this paper is of decent quality, but in my opinion not a competitive paper at ICLR. Also, I couldn’t find some of the promised changes in the revised version of the paper, and feel like the authors could have spent more effort to carefully include the reviewer's feedback in the main paper. Most of the changes that have been made were added to the Appendix, sometimes without even referencing/mentioning them in the main paper (e.g. Section G of the Appendix is unreferenced in the main paper).

---

> > ### Author Response · Authors · 2024-11-29
> > **Response to Reviewer teJM #1**
> >
> > We sincerely appreciate the reviewer’s thorough review of our responses and are pleased that we were able to address the majority of the concerns (11/17). Below are our detailed responses to the remaining comments.
> >
> > **Q1: I think the author’s response in the rebuttal is nice and should be worked into the paper.**
> >
> > A1: We are delighted that our rebuttal has met your satisfaction. As the main text cannot be modified at this stage, we will include the citation in the final published version of the paper.
> >
> > **Q2: I think this evidence nicely supports the claim, but I wish the authors had referred to it earlier, possibly in Section 3.4 when they introduce their multimodal regularization strategy. I still think this is a very nice observation that deserves more attention.**
> >
> > A2: Thank you for recognizing the evidence and explanation we provided. The evidence was demonstrated through specific experiments, which is why it was included in ablation study (Section 5.4). We also believe that this finding is significant and has the potential to inspire future research. Thank you for your suggestion to better highlight this contribution. We will relocate the experimental table to Section 3.4 in the final revised version of the manuscript.
> >
> > **Q3:  Unfortunately, I can’t find Section A1 in the revised version of the supplementary material.**
> >
> > A3: As mentioned in our response to A1, we have already addressed your concern regarding Q3. We apologize for any confusion caused. To clarify, here is a restatement of the original Q3 question. *”Authors claim multiple times (for example in L028, L104, and L536) that they are the first to use 3DGS to model RGB and thermal images. This statement is actually not true and should be revised. The authors of [5] already presented a 3DGS-based method that incorporates RGB and thermal data (see Section 4.5 of the main paper and Section 4 of supplementary).“*
> >
> > Next, we outline the similarities and differences between our work and [5].
> >
> > **[5] is also a NeRF-based method**, focusing on NeRF-based RGB reconstruction under low-light conditions. Similar to our work, [5] demonstrates that thermal modalities can enhance RGB reconstruction quality, particularly in low-light scenarios. However, [5] largely overlooks the importance of reconstructing the thermal modality itself. In contrast, our work not only improves RGB reconstruction in low-light conditions but also reconstructs a thermal field that is well-suited for thermal analysis.
> >
> > **Q4:  I don’t see the point of not using the dataset proposed in [3]. If it is of bad quality, then all methods evaluated on this dataset simply perform equally badly, but the comparison across methods would still be fair according to my understanding.**
> >
> > A4: [3] lacks multi-view consistency, meaning that temperature measurements for the same object vary across different views. This inconsistency causes the ground truth (GT) temperature measurements during testing to differ from those used during training for the same location. In the context of 3D reconstruction, this implies that the GT itself is unreliable. Comparing results on a dataset where the GT is likely inaccurate makes it difficult to draw meaningful conclusions. The differences in our respective fields may have caused some confusion. In the 3D reconstruction domain, multi-view consistency is critical as it directly impacts the accuracy of the GT.
> >
> > **Q8: I was actually referring to a side-by-side comparison of thermal renderings (i.e., images) with and without the smoothness term. I apologize in case I wasn’t clear enough.**
> >
> > A8: Apologies for the misunderstanding. We have provided quantitative comparisons, and we can certainly include some qualitative comparisons as well. We will include relevant images in the supplementary materials of the final version of the paper.

---

> > > ### Author Response · Authors · 2024-11-29
> > > **Response to Reviewer teJM #2**
> > >
> > > **Q17：  Thanks for the clarification. As this is very important information, I’m wondering why the authors did not include it in the revised version of the paper ...**
> > >
> > > A17：Thank you for acknowledging our response in A17. We are glad to continue the discussion with you. Since alumina is the most common default material for standard checkerboard calibration boards, we overlooked mentioning this in our previous revision. To ensure clarity for all readers, we will specify the material in Section 3.2 of the revised version.
> > >
> > > Regarding calibration, precise calculation of camera parameters is essential. Due to the high precision required for calibration, the perforated aluminum plate must have each hole's radius strictly controlled (to at least millimeter accuracy), as well as precise and consistent distances between the centers of the holes. We believe manufacturing such an aluminum plate is not a simple task, and we have not come across any commercially available versions. On the other hand, standard checkerboard calibration boards can be easily purchased online for as little as $5, offering both convenience and affordability. Additionally, these boards are compatible with the most commonly used color cameras, whereas an aluminum plate with holes is not suitable for calibrating color cameras. Standard checkerboard boards are typically essential equipment for precise camera calibration, especially for teams working with robotics.
> > >
> > > **Q18: Most of the changes that have been made were added to the Appendix, sometimes without even referencing/mentioning them in the main paper (e.g. Section G of the Appendix is unreferenced in the main paper).**
> > >
> > > A18:Thank you for your valuable feedback. Due to the page limitations for the conference, we were unable to include everything in the main text as we would in a journal submission. We have tried to prioritize the most important content in the main paper, which is why detailed dataset descriptions, some discussions, and additional experiments are placed in the supplementary materials.
> > >
> > > We appreciate your suggestions to improve the quality of the paper. In the revised version, we will ensure that all supplementary materials are referenced in the main paper.
> > >
> > > Finally, we hope our response helps to address your concerns.

---

> > > > ### Comment · Reviewer_teJM · 2024-12-02
> > > >
> > > > Thank you again for the detailed response.
> > > >
> > > > A3: Thanks for pointing me to response A1 again. My apologies, I don't know why but I was somehow looking for Appendix A1. This concern has been, in fact, already resolved with the author's previous response.
> > > >
> > > > A17: I see why it might be difficult to exactly locate the holes' midpoints when using a perforated aluminum plate. This is indeed an advantage of a standard checkerboard pattern.
> > > >
> > > > Overall, I appreciate the author's responses. Since I don't have any critical concerns at this stage and do want to recognize the author's effort in providing a thorough rebuttal, I am willing to raise my score. However, I still don't think this is a competitive paper at ICLR; the technical contribution is, from my perspective, rather limited. I would also encourage authors to carefully revise the paper according to the feedback received during this rebuttal.

---

> > > > > ### Author Response · Authors · 2024-12-02
> > > > > **Response to Reviewer teJM**
> > > > >
> > > > > Thank you for recognizing our responses and for improving the final score. We will further refine our paper based on your suggestions.

---

### Official Review · Reviewer_B1kF · 2024-11-04

**Soundness:** 3
**Presentation:** 3
**Contribution:** 4
**Rating:** 8
**Confidence:** 4

**Summary:**

This paper introduce ThermalGaussian, a multimodal Gaussian technique that renders high-quality RGB & thermal images from new views.
Also they introduce a new real-world dataset named RGBT-Scenes, to address the issue of existing problems.
Authors propose multimodal regularization to prevent overfitting and show that ThermalGaussian outperforms previous methods by enhancing rendering quality, reducing storage requirements, and improving efficiency.

**Strengths:**

The proposed approach is novel, especially in integrating thermal and RGB images in 3D reconstruction via Gaussian splatting. And the experimental results are clear, and include diverse evalutaion metrics and ablation studies to validate performance of this methods.

Also, introducing the RGBT-Scenes dataset is a valuable contribution that can serve as a benchmark for future work in thermal 3D reconstruction.

**Weaknesses:**

The process of multimodal initialization and the specific configurations for the thermal Gaussian construction are intricate, making the approach challenging to implement without in-depth knowledge. And, although the RGBT-Scenes dataset is comprehensive, details regarding environmental diversity and lighting conditions could strengthen its applicability across broader contexts.

**Questions:**

- Can the authors clarify the choice of hyperparameters for multimodal regularization? Are there general guidelines for adjusting them?

- How does the model handle extreme cases, such as highly reflective surfaces or objects at temperatures that differ significantly from ambient conditions?

- Could the RGBT-Scenes dataset be expanded to include data for real-time applications, like moving objects or dynamic scenes?

**Details Of Ethics Concerns:**

No ethical concerns identified.

---

> ### Author Response · Authors · 2024-11-25
> **Response to Reviewer  B1kF**
>
> We sincerely appreciate the positive feedback and valuable suggestions provided by the reviewer. Here are our responses to your comments.
>
> **Q1: The process of multimodal initialization and the specific configurations for the thermal Gaussian construction are intricate, making the approach challenging to implement without in-depth knowledge.**
>
> A1: To facilitate the implementation, we will release our code upon acceptance. This will enable more researchers to directly utilize the methodology presented in our paper. The multimodal initialization process mainly focuses on registering various types of cameras. Although mastering this process may take some time for beginners, it is a crucial skill for researchers in fields like robotics and computer vision who wish to effectively utilize images captured by various cameras.
>
> **Q2: Although the RGBT-Scenes dataset is comprehensive, details regarding environmental diversity and lighting conditions could strengthen its applicability across broader contexts.**
>
> A2: We sincerely appreciate the reviewer's acknowledgment of the comprehensiveness of our RGBT-Scenes dataset. To ensure diversity, we include both forward-facing and 360-degree scenes, encompassing indoor and outdoor environments.  Based on your recommendations, we have collected additional data, including two low-light scenes, two complex scenes, and a scene featuring special glass materials. We will release all the data upon acceptance. The detailed description of the dataset is presented in **Appendix A**.
>
> **Q3: Can the authors clarify the choice of hyperparameters for multimodal regularization? Are there general guidelines for adjusting them?**
>
> A3: For multimodal regularization, we empirically set the hyperparameters: 0.3 for the color modality and 0.6 for the thermal modality.  The lower texture details in thermal images lead to a loss that is typically half of that observed in the color modality. Therefore, we set the hyperparameter for the thermal modality to twice that of the color modality during multimodal regularization.
>
>
> **Q4: How does the model handle extreme cases, such as highly reflective surfaces or objects at temperatures that differ significantly from ambient conditions?**
>
> A4: Some extreme cases pose significant challenges for achieving successful 3D reconstruction. We are actively gathering additional datasets for extreme cases. Addressing these extreme cases is crucial for our future work.
>
> **Q5: Could the RGBT-Scenes dataset be expanded to include data for real-time applications, like moving objects or dynamic scenes?**
>
> A5: This comment is very valuable. We are currently conducting research on dynamic thermal-RGB reconstruction and have collected datasets for more than ten dynamic scenes. These scenes include more diverse settings, lighting variations, and scenarios with significant temperature differences. We showcase some dynamic scenes in **Appendix A**. In our upcoming work, we will also release the RGBT-Dynamic-Scenes dataset.

---

### Author Response · Authors · 2024-11-25
**General Comment**

We thank all the reviewers for their detailed feedback and insightful questions.  We are glad to hear that the reviewers acknowledged the contributions of this paper. We have carefully revised our paper according to the suggestions and questions from the reviewers (marked in blue). We have provided detailed responses to each reviewer's questions below.

---

### Meta-Review · Area_Chair_SaoZ · 2024-12-20

**Metareview:**

The paper introduces ThermalGaussian, a method that extends 3D Gaussian Splatting (3DGS) for joint RGB and thermal 3D scene reconstruction. It proposes three multimodal training strategies, thermal-specific regularization techniques, and a new dataset (RGBT scenes) to achieve improved RGB and thermal rendering quality while reducing storage costs by ~90%.

The strengths of the work are its extension of 3D Gaussian Splatting for joint RGB and thermal 3D reconstruction, introducing multimodal training strategies and thermal-specific regularization to improve rendering quality and efficiency. It significantly reduces storage costs while improving RGB and thermal image performance, especially in challenging scenarios such as low light. The newly introduced RGBT scenes dataset is a valuable resource for the community.

Reviewers found that the comparisons with recent methods like Thermal-NeRF and datasets such as ThermalMix is needed. Also the paper does not sufficiently address the practical limitations of the proposed method, such as challenges in real-world data alignment, and provides limited discussion of the dataset characteristics and applicability.

**Additional Comments On Reviewer Discussion:**

During the rebuttal period, the authors provided answers to the reviewers' questions thoroughly. The authors added an additional experiment on ThermoNeRF, and several clarifications regarding the reviewer's comments.
After rebuttal, most reviewers agree to accept the paper.

---

### Decision · Program_Chairs · 2025-01-22

Accept (Poster)